# Time-course changes in mental distress and their predictors in response to the coronavirus disease 2019 (COVID-19) pandemic: A longitudinal multi-site study of hospital staff

**Yosuke Kameno**[1,2], **Tomoko Nishimura**[3], **Yumi Naito**[2], **Daisuke Asai**[1,2], **Jun Inoue**[4], **Yosuke Mochizuki**[1], **Tomoyo Isobe**[1], **Atsuko Hanada**[5], **Noriyuki Enomoto**[2], **Hidenori Yamasue**[1,2]*

1 Department of Psychiatry, Hamamatsu University School of Medicine, Hamamatsu, Shizuoka, Japan,
2 Health Administration Center, Hamamatsu University School of Medicine, Hamamatsu, Shizuoka, Japan,
3 Research Center for Child Mental Development, Hamamatsu University School of Medicine, Hamamatsu, Shizuoka, Japan, 4 Department of Child and Adolescent Psychiatry, Hamamatsu University School of Medicine, Hamamatsu, Shizuoka, Japan, 5 Department of Nursing, Hamamatsu University Hospital, Hamamatsu, Shizuoka, Japan

* yamasue@hama-med.ac.jp

**Data Availability Statement:** All relevant data are within the paper and its Supporting Information files.

## Abstract

The coronavirus disease 2019 (COVID-19) pandemic provides a unique opportunity studying individual differences in the trajectory of mental distress to relatively homogeneous stressors by longitudinally examining time-course changes between pandemic waves. For 21 months, we tested the effects of COVID-19 waves on mental health among 545 staffs at 18 hospitals treating COVID-19 patients in Shizuoka Prefecture, Japan. Contrary to increasing new infected cases as waves progressed, initially elevated psychological distress (K6) and fear of COVID-19 (FCV-19S) were decreased among waves (K6: B = -.02, 95% confidence interval [CI] = -.03 to -.01; FCV-19S: B = -.10, 95% CI = -.16 to -.04). This initial increase and subsequent decrease in K6 and FCV-19S were more prominent in individuals with high trait anxiety (K6: B = 1.55, 95% CI = 1.18 to 1.91; FCV-19S: B = 4.27, 95% CI = 2.50 to 6.04) and in occupations other than physicians or nurses. The current study revealed time-course changes in psychological distress and fear regarding COVID-19 in each pandemic wave and across waves, and indicated the usefulness of trait anxiety and occupation as predictors of mental health outcomes.

## Introduction

The successive waves of the coronavirus disease 2019 (COVID-19) pandemic provide a unique opportunity to test the effects of repeated stressors on time-course changes in mental health. Previous research on the effects of stressors on mental health has been hampered by the nature

**Funding:** This research was supported by the disability care division of the Shizuoka prefectural government, funding awarded to HY. The funding source had no role in the design of the study; collection, analysis, and interpretation of the data; preparation, review, or approval of the manuscript; and decision to submit the manuscript for publication.

**Competing interests:** The authors have declared that no competing interests exist.

of macro-stressors, particularly their unpredictable and asynchronous occurrence and their variability between individuals. However, the ongoing COVID-19 pandemic represents a global, relatively synchronously starting and homogeneous set of stressors, allowing us to study a large number of affected individuals simultaneously. Furthermore, unlike other group- or age-specific stressors, the pandemic affects people of all societal groups and all ages, offering the possibility of exploring predictors of mental health outcomes [1]. Thus, the repeated pandemic waves provide a unique opportunity to explore time-course changes and their predictors in relation to mental distress in response to a relatively uniform stressor in a large number of affected individuals simultaneously.

A number of previous cross-sectional studies reported associations between various factors and individual differences in responses to the COVID-19 pandemic, including female gender, working in healthcare, age, trait anxiety, social class, and governmental stringency [1–3]. Regarding occupation, a number of studies reported heightened mental distress in hospital staff, such as frontline nurses [4–6]. These individual differences in mental distress in response to the pandemic may be useful predictors of mental health outcomes.

A number of studies reported increased mental distress in response to the pandemic during the first several weeks or months [3,7]. However, longitudinal studies will be needed to clarify the time-course changes between repeated pandemic waves as well as those during each wave.

To investigate time-course changes and their individual differences in mental distress under the ongoing COVID-19 pandemic, the current multisite longitudinal study examined participants with high homogeneity in a sample of hospital staff of working age, most of whom were Japanese and living in the same prefecture under the same level of governmental stringency. The homogeneity of this sample enhanced our ability to clarify the relationships between individual differences in characteristics among the participants (such as occupation type among hospital staff and trait anxiety) and mental distress during the pandemic. Mental distress was indexed by psychological distress and fear of COVID-19.

## Methods

### Study design and participants

We conducted an online survey of hospital staff who worked in hospitals that treated patients with COVID-19 infection in Shizuoka Prefecture in Japan. The survey was contracted to us by the disability care division of the prefectural government, with the aim of longitudinally investigating and monitoring the psychological burden on medical staff. The survey was conducted from 1 February 2021 to 31 October 2022; this period covered the 3rd to 7th waves of the COVID-19 pandemic in Japan. Although a state of emergency was declared nationwide, a full lockdown was not legally enforced in Japan. The government of Shizuoka Prefecture asked citizens to avoid both non-essential outings and travel to another prefecture, and requested that they take basic precautionary measures (washing hands, using masks effectively, avoiding closed spaces with poor ventilation, avoiding crowded places with many people nearby, and avoiding close-contact settings, e.g., close-range conversations within 2m). These recommendations did not change significantly during the survey period. Many citizens followed these recommendations. Before starting the survey, we prepared questionnaire forms and a cloud database to store the responses. We repeatedly held online meetings and explained how to access and answer the forms. We recommended this system to all 31 general and some long-term care hospitals in the prefecture that treated patients with COVID-19. The participants were registered in a database when they answered the questionnaire for the first time, and received automated e-mails once a month to remind them to answer the questionnaire. In order to reduce selection bias, the staff in charge of this survey in each hospital widely called

the survey to the attention of medical staff via handouts and posters that asked them to the questionnaire. After the survey period, we checked all responses to make sure that the same individual did not answer more than once per day. If the hospital name, department name, and email account on two responses matched, we regarded them as having been completed by the same individual. If we found duplicate responses, we considered the last one to be the correct one and deleted the others. After finishing the survey, we obtained ethical approval and analyzed the survey results. Therefore, this study was designed as a retrospective one.

## Outcome measures in longitudinal assessments

To longitudinally evaluate the mental distress in relation to the pandemic, psychological distress and fear in response to COVID-19 were assessed using the Kessler Psychological Distress Scale (K6) [8] and Fear of COVID-19 Scale (FCV-19S) [9] respectively, because questionnaires with fewer items were suitable in this survey of busy hospital staff during the pandemic, on the basis of our experience in a preliminary study [10]. We used the Japanese version of the K6 [8], a standard six-item instrument for primary care and research, with a five-point Likert scale ranging from 0 ("none of the time") to 4 ("all of the time"). The total score has been used as an indicator of mental distress (over 5 points) or mood and anxiety disorders equivalent (over 10 points) [11–13]. A previous study reported that Cronbach's α of the Japanese version of K6 scale was 0.85 [11]. The screening performance of this scale was comparable with that of Center for Epidemiologic Studies-Depression Scale (CES-D), and better than that of Depression and Suicide Screen (DSS) in the Japanese version [11]. Fear of COVID-19 was measured using the Japanese version of the FCV-19S [9], a seven-item instrument developed during the COVID-19 pandemic, with a five-point Likert response scale ranging from 1 ("strongly disagree") to 5 ("strongly agree") [14]. Scores on the FCV-19S were reported to be positively correlated with anxiety and depression scale scores [14]. A previous study also revealed Cronbach's α of the Japanese version of FCV-19S scale was 0.82 [9]. This scale showed a significant positive correlation to Generalized Anxiety Disorder (GAD)-7, Patient Health Questionnaire for Adolescents (PHQ-A), and Perceived Vulnerability to Disease Scale (PVDS) in the Japanese version. The scale had sufficiently high internal confidence and moderately good construct validity [9].

## Demographic assessments at baseline

When participants accessed the questionnaire for the first time, they answered questions regarding the following demographic factors: age, gender, and occupation. As a potential predictor of mental distress, we hypothesized that individuals with high anxiety trait would be more likely to feel strongly stressed when treating patients with COVID-19, and to exhibit mental distress. In addition, a previous cross-sectional study showed that trait anxiety was associated with perceived vulnerability concerning health and fear of COVID-19 [15]. Therefore, participants were checked for levels of trait anxiety using State-Trait Anxiety Inventory (STAI) [16,17]. We used the Japanese version of the STAI [16], a well-validated 40-item tool used to evaluate the severity of anxiety symptoms in clinical practice and research. The STAI questionnaire consisted of state and trait anxiety and contained 20 items for each type of anxiety. The total scores were obtained by summing the values assigned to each item rated from 1 to 4 points, with higher scores suggesting higher levels of anxiety. The STAI classified anxiety into five stages using the total score: stage I reflected very low scores, stage II reflected low scores, stage III reflected normal scores, stage IV reflected high scores, and stage V reflected very high scores [16]. In addition, participants were asked each time whether they had experienced direct contact with patients infected with COVID-19.

## Ethical considerations

The ethical committee of Hamamatsu University School of Medicine approved this study protocol as a retrospective study on 29 March 2022 and informed consent was obtained from participants in the form of an opportunity to opt out on the website (20–083).

## Statistical analysis

First, a generalized linear mixed model (GLMM) was used to examine longitudinal changes in psychological distress and fear of COVID-19 during the observation period; additionally, it was used to determine whether the amount of change differed according to individual attributes. We focused on individual trait anxiety because a previous cross-sectional study showed that trait anxiety was associated with perceived vulnerability concerning health and fear of COVID-19 [15]. Assuming that the amount of change differs depending on the levels of trait anxiety, we classified trait anxiety into the following three categories: below normal (stage I–III), high (stage IV), and very high (stage V) [16]. We also focused on the occupation type, because the prevalence of mental health problems was reported to be high among healthcare workers, especially among nurses, during the COVID-19 pandemic [18,19]. Furthermore, the associations between direct contact with infected patients and psychological distress and fear of COVID-19 were examined in those participants who had direct contact with infected patients.

To depict the trajectories of psychological distress and fear of COVID-19, we included participants who took at least two measurements during the observation period. In the analyses, both the intercept and slope of the individual regression line were assumed to differ (random intercept and random slope model). In Model 0, we estimated a model that included only the outcome variable (K6 or FCV-19S score) to calculate the within-individual intra-class correlation coefficient (ICC). In Model 1, to examine the linear trends in change in outcome variables, the effects of time, individual trait anxiety, and occupation type (doctor, nurse, and other hospital staff) were included in the model. Because previous studies indicated that younger age and female gender were associated with a higher likelihood of developing psychological distress [3,20], age and gender were added to the model as individual background characteristics. In Model 2, we included interaction terms between time and individual trait anxiety and between time and occupation type in the model to examine whether the change trends differed by trait anxiety and occupation type. Regarding the association between direct contact with infected patients and psychological distress and fear of COVID-19, whether there was direct contact or not was not fixed by the participant, but differed at the time of the response, so it was entered into the GLMM as a time-varying variable, without including an interaction term with time. Individual background characteristics were also included in Model 2. A significance level of $\alpha = .05$ was used for interpreting the statistical significance of the results.

In Japan, the number of newly infected cases increased rapidly after the beginning of 2022, exceeding 100,000 cases; however, no state of emergency was announced. Considering the possibility that psychological states may differ between the first and second halves of the observation period, the analyses were also stratified by the observation period (before and after 100 weeks [December 2021] from the beginning of the COVID-19 outbreak in Japan).

Additionally, the observation period was divided into two periods (before and after 100 weeks), and a GLMM was then conducted for each period. This division was employed because a) the number of newly infected cases differed considerably and b) policies for infection control changed between the first and second half of the observation period, which could have affected mental health differently. Multiple comparisons were corrected using Bonferroni's method. GLMM was conducted using STATA version 17.0 [21].

## Results

### Study participants and the demographics at the initial assessment

Eighteen of the 31 contacted hospitals participated in the survey, and 545 participants at these 18 hospitals who answered the K6 and FCV-19S were analyzed in this study. The total number of responses was 1,381. The mean number of responses per person was 2.53 (standard deviation [SD] 3.09, range 1–20) and the mean interval of responses was 57.94 days (SD 64.29, range 1–450).

The mean K6 score at the initial assessment was 6.01 (SD 5.29, range 0–24), which was higher than the pre-pandemic scores of 3.31–4.79 reported in previous studies [22–24]. Over 5 points of K6 indicated psychological distress and over 10 points mood and anxiety disorder equivalent. In our study, the number of participants scoring 5 and 10 points or higher was 289 (53.03%) and 128 (23.49%), respectively. This prevalence was higher than that of the general situation before the pandemic in Japan (5 points 28.9%, 10 points 10.2%) [23].

The mean FCV-19S score at the initial assessment was 19.23 (SD 5.52, range 7–35). Previous studies reported that the mean FCV-19S score in response to the pandemic among healthcare workers was 12.89–20.02 [25–32]. The current result was similar to the findings of these previous studies.

Of the 545 individuals who answered the K6 and FCV-19S, 426 participants reported the demographics at the initial assessment including gender (male: n = 105, 24.65%; female: n = 321, 75.35%), age (mean = 37.94 years old, SD = 12.41, range: 20–74), and occupation. Participants' occupations were as follows: physician (n = 36, 8.45%), nurse (n = 249, 58.45%), public health nurse (n = 2, 0.47%), pharmacist (n = 4, 0.94%), psychologist (n = 7, 1.64%), radiological technician (n = 6, 1.41%), laboratory technician (n = 32, 7.51%), care manager (n = 2, 0.47%), social worker (n = 1, 0.23%), psychiatric social worker (n = 1, 0.23%), speech-language therapist (n = 5, 1.17%), occupational therapist (n = 19, 4.46%), physical therapist (n = 19, 4.46%), caregiver (n = 14, 3.29%), dietitian (n = 3, 0.70%), nursery teacher (n = 1, 0.23%), and office worker (n = 25, 5.87%).

The mean score of STAI-trait score was 52.78 (SD 10.88, range 20–79). High scores for STAI-trait (over 44 in men, 45 in women) was defined as severe anxiety [15]. Therefore, many participants had severe anxiety during repetitive pandemic waves (STAI-trait: 335 of 426 (78.64%) participants) (Table 1).

### Changes in psychological distress and fear of COVID-19

For K6 scores (psychological distress), a GLMM that assumed a negative binomial distribution was performed because these scores were not normally distributed. The ICC was .73, indicating a strong correlation among observations within the same individual. Table 2 shows the effects (and their 95% confidence intervals [CIs]) of time, trait anxiety, and occupation type on K6 scores for the total observation period. In Model 1, the effect of time was significantly negative, indicating that K6 scores decreased during these periods (S1A Fig). The effect of trait anxiety was significantly positive for both the high and very high trait anxiety groups at all observation periods, indicating higher K6 scores compared with the below-average trait anxiety group. However, interaction between time and trait anxiety, between time and occupation type, and main effects of them were not significant in Model 2. In the additional analyses, the interaction between time and trait anxiety was significant for the high and very high trait anxiety group after 100 weeks, indicating a significant decrease in K6 scores in this period compared with the below-average trait anxiety group (S1 Table, Fig 1A). The interaction between time and occupation type was significant for nurses after 100 weeks, and nurses had lower but increasing K6 scores compared with other occupations (S1 Table, Fig 2A).

**Table 1. Demographic characteristics and clinical measures at baseline.**

| | Number of response | n (%) | Mean | S.D. | Min | Max |
|---|---|---|---|---|---|---|
| Outcome measures at the initial assessment | | | | | | |
| K6 | 545 | | 6.01 | 5.29 | 0 | 24 |
| Over 5 points | | 289 (53.03%) | | | | |
| Over 10 points | | 128 (23.49%) | | | | |
| FCV-19S | 545 | | 19.23 | 5.52 | 7 | 35 |
| Demographics at the initial assessment | | | | | | |
| Gender | 426 | | | | | |
| Male | | 105 (24.65%) | | | | |
| Female | | 321 (75.35%) | | | | |
| Age | 425 | | 37.94 | 12.41 | 20 | 74 |
| Occupation | 426 | | | | | |
| Doctor | | 36 (8.45%) | | | | |
| Nurse | | 249 (58.45%) | | | | |
| Public Health Nurse | | 2 (0.47%) | | | | |
| Pharmacist | | 4 (0.94%) | | | | |
| Psychologist | | 7 (1.64%) | | | | |
| Radiological Technician | | 6 (1.41%) | | | | |
| Laboratory Technician | | 32 (7.51%) | | | | |
| Care Manager | | 2 (0.47%) | | | | |
| Social Worker | | 1 (0.23%) | | | | |
| Psychiatric Social Worker | | 1 (0.23%) | | | | |
| Speech-Language Therapist | | 5 (1.17%) | | | | |
| Occupational Therapist | | 19 (4.46%) | | | | |
| Physical Therapist | | 19 (4.46%) | | | | |
| Carer | | 14 (3.29%) | | | | |
| Dietitian | | 3 (0.70%) | | | | |
| Nursery Teacher | | 1 (0.23%) | | | | |
| Office Worker | | 25 (5.87%) | | | | |
| STAI trait anxiety | 426 | | 52.78 | 10.88 | 20 | 79 |
| Stage I | | 2 (0.47%) | | | | |
| Stage II | | 15 (3.52%) | | | | |
| Stage III | | 74 (17.37%) | | | | |
| Stage IV | | 131 (30.75%) | | | | |
| Stage V | | 204 (47.89%) | | | | |

Abbreviations: S.D., standard deviation.

For FCV-19S scores (fears of COVID-19), a GLMM assuming a Gaussian (normal) distribution was performed. The ICC was .68, indicating a high correlation among observations within the same individual. Table 3 shows the effects and 95% CI of time, trait anxiety, and occupation type on FCV-19S scores for total observation period. As with K6 scores, in Model 1, the effect of time was significantly negative, indicating a decrease in FCV-19S scores during the observation periods (S1B Fig). The effect of trait anxiety was positive for very high trait anxiety groups. In Model 2, a significant interaction between time and occupation type and a significant main effect of occupation type indicated that doctors had lower but increasing FCV-19S scores compared with other hospital staff for the total observation period (Fig 2B). In

**Table 2. Effects of time and trait anxiety on K6 scores during the total observation period.**

| | Model 1 | Model 2 |
|---|---|---|
| | B (95% CI) | B (95% CI) |
| Time (month) | -.02 (-.03, -.01)* | -.02 (-.07, .02) |
| Trait anxiety (below average group is reference) | | |
| High | .82 (.45, 1.19)* | 6.37 (-24.42, 37.16) |
| Very high | 1.55 (1.18, 1.91)* | 18.18 (-12.59, 48.94) |
| Time×Trait anxiety | | |
| High | | -.01 (-.05, .03) |
| Very high | | -.02 (-.06, .02) |
| Occupation type (other hospital staff is reference) | | |
| Doctor | -.26 (-.81, .30) | -17.26 (-50.42, 15.91) |
| Nurse | -.22 (-.52, .08) | -16.91 (-38.83, 5.00) |
| Time×Occupation type | | |
| Doctor | | .02 (-.02, .07) |
| Nurse | | .02 (-.01, .05) |
| Background characteristics | | |
| Age | -.001 (-.01, .01) | -.001 (-.01, .01) |
| Sex (female) | .52 (.15, .89)* | .55 (.18, .93)* |

*Statistical significance after Bonferroni correction.

Abbreviations: B, regression coefficient; 95% CI, 95% confidence interval.

the additional analyses, the interaction between time and trait anxiety was significant only for the very high trait anxiety group for 100 weeks or earlier, indicating that FCV-19S scores in this group decreased significantly in this period compared with the below-average trait anxiety group (S2 Table). The interaction between time and occupation type was also significant only for the nurse, indicating that nurses had lower but increasing FCV-19S scores compared with other hospital staff for the 100 weeks or earlier period (S2 Table).

As for direct contact with patients infected with COVID-19, 451 participants who had direct contact with infected patients were included. Such contact was not associated with psychological distress (B = .01, 95% CI: -.11 to .14), but was significantly associated with fear of COVID-19 (B = .72, 95% CI: .05 to 1.39).

## Discussion

The current longitudinal study tested the effects of the repeated waves of the COVID-19 pandemic (a relatively homogeneous stressor) on mental health of hospital staff living in the same prefecture in Japan (a relatively homogeneous population). Several main findings were revealed, as follows. Although the number of infected people was increased during the successive waves of the pandemic, both the initially elevated psychological distress and fear of COVID-19 among hospital staff were decreased from the 3rd to the 7th waves of the pandemic. This initial increase and subsequent decrease in psychological distress and fear of COVID-19 was observed in a more pronounced way in individuals with high trait anxiety during the study period covering the 3rd and 4th waves. In contrast to other hospital staff, both the initial increase and subsequent decrease in fear regarding COVID-19 was less prominent among nurses and physicians.

Consistent with previous studies showing elevated psychological distress and fear regarding COVID-19 among hospital staff during the COVID-19 pandemic [4–6], the current study

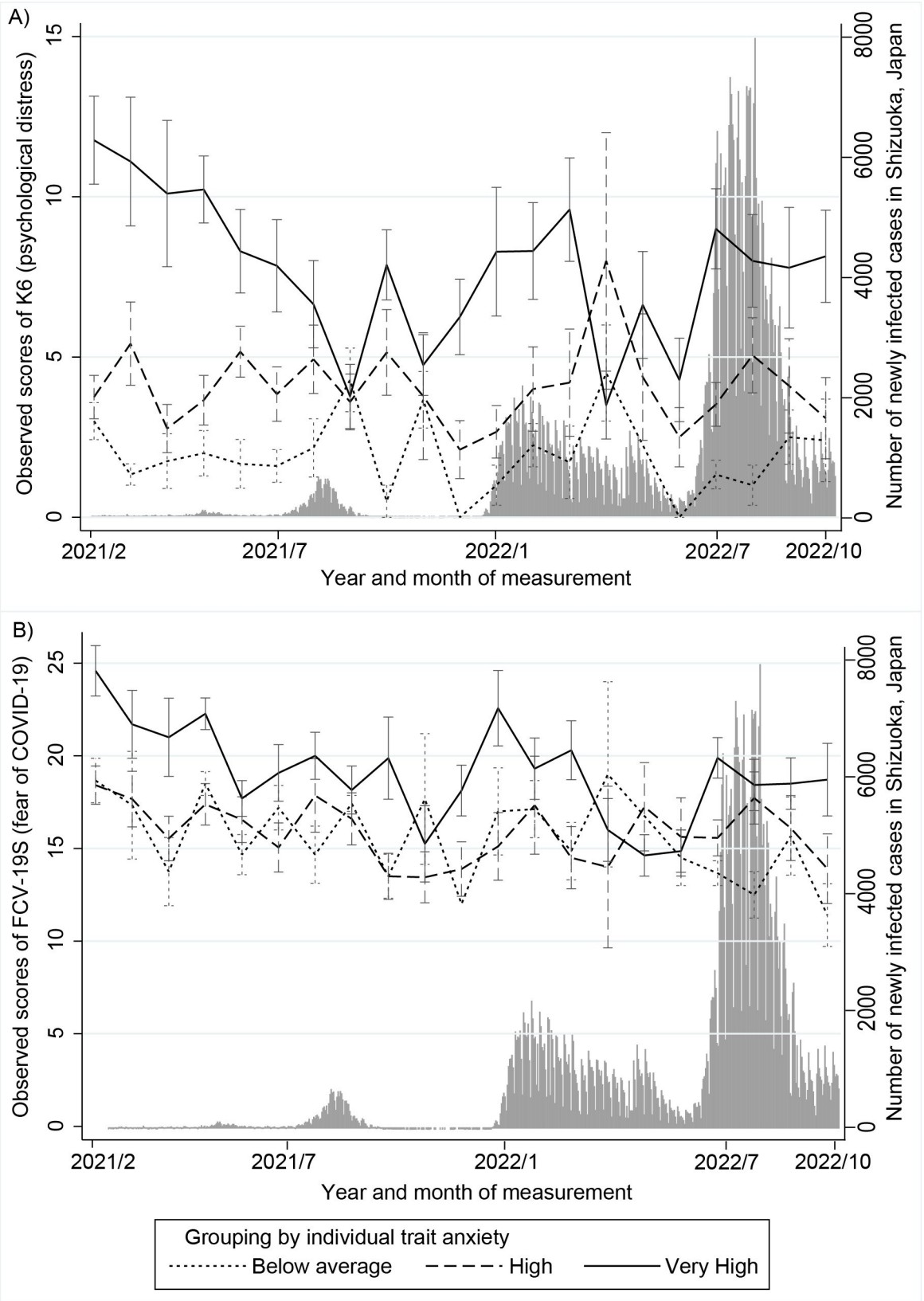

**Fig 1. Changes in K6 and FCV-19S scores by level of trait anxiety and number of newly infected cases.** Changes in K6 (A) or FCV-19S (B) scores were shown by level of trait anxiety and number of newly infected cases during the observational period. Error bar indicates standard error of the mean (SEM).

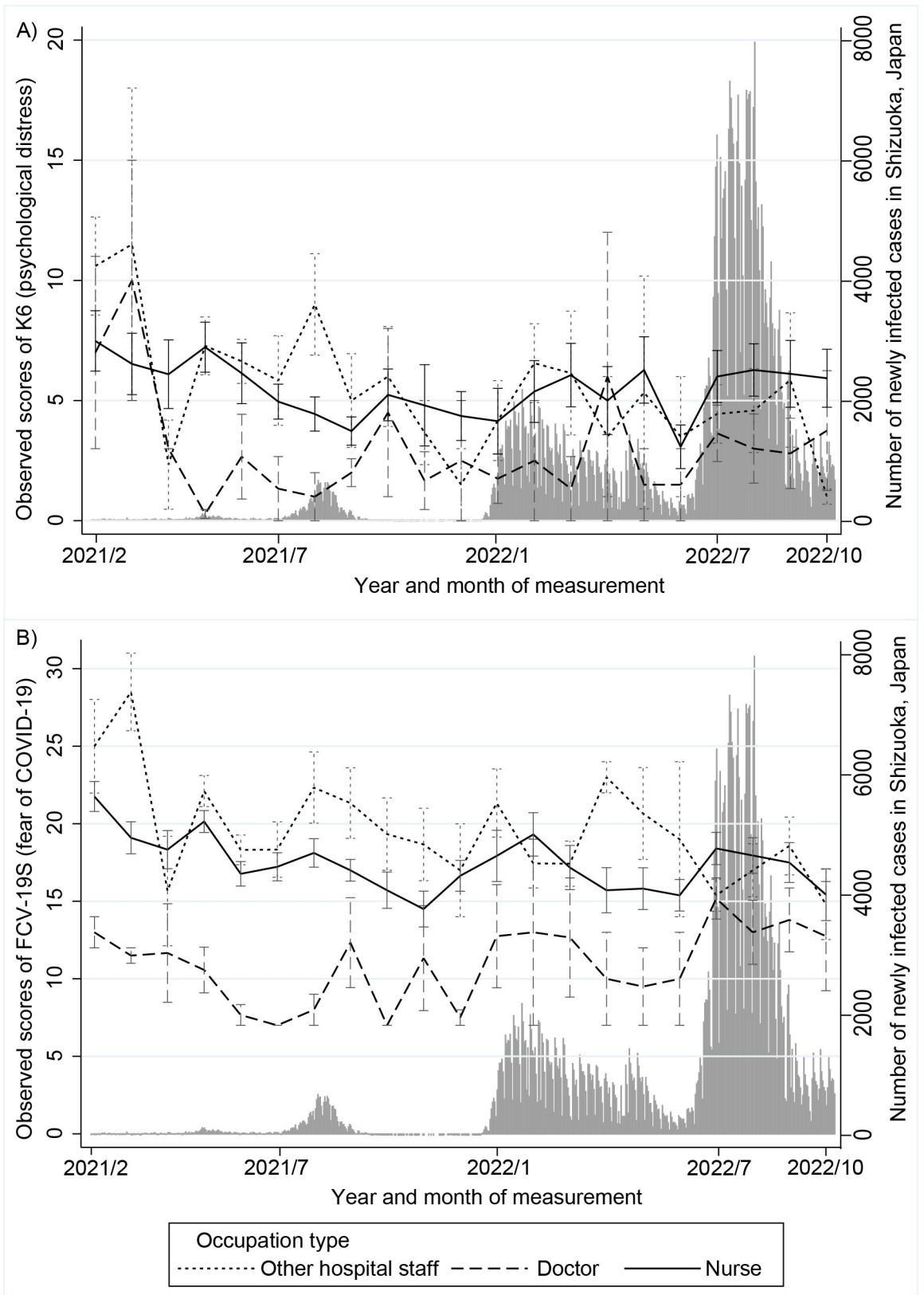

**Fig 2. Changes in K6 and FCV-19S scores by occupation type and number of newly infected cases.** Changes in K6 (A) or FCV-19S (B) scores were shown by occupation type and number of newly infected cases during the observational period. Error bar indicates SEM.

**Table 3. Effects of time and trait anxiety on FCV-19S scores during the total observation period.**

| | Model 1 | Model 2 |
|---|---|---|
| | B (95% CI) | B (95% CI) |
| Time (month) | -.10 (-.16, -.04)* | -.17 (-.36, .02) |
| Trait anxiety (below average group is reference) | | |
| High | 1.87 (.10, 3.64) | 41.9 (-90.2, 174.1) |
| Very high | 4.27 (2.50, 6.04)* | 92.5 (-42.3, 227.2) |
| Time×Trait anxiety | | |
| High | | -.05 (-.23, .12) |
| Very high | | -.12 (-.30, .06) |
| Occupation type (other hospital staff is reference) | | |
| Doctor | -2.68 (-5.42, .05) | -268.9 (-420.8, -117.0)* |
| Nurse | -1.73 (-3.29, -.16) | -98.6 (-210.1, 12.9) |
| Time×Occupation type | | |
| Doctor | | .36 (.15, .56)* |
| Nurse | | .13 (-.02, .28) |
| Background characteristics | | |
| Age | -.03 (-.09, .03) | -.03 (-.09, .03) |
| Sex (female) | 3.22 (1.36, 5.08)* | 3.33 (1.49, 5.16)* |

*Statistical significance after Bonferroni correction.

Abbreviations: B, regression coefficient; 95% CI, 95% confidence interval.

participants also showed elevated psychological distress and fear of COVID-19 during the pandemic at the start of the investigation. It is possible that the psychometric scores were high because of the generally high amount of uncertainty and fear of COVID-19 during the early stages of the pandemic. However, the elevated psychological distress and fear of COVID-19 were decreased during successive waves. Considering the increase in the number of infected people during successive waves of the pandemic, decreases in both elevated psychological distress and fear of COVID-19 during the successive waves were notable. In accord with the current findings, a few previous studies have reported decreases in depressive symptoms and anxiety between the pandemic waves during shorter periods (i.e., 8 to 11 months) [33,34]. These decreases in anxiety or fear regarding COVID-19 are understandable considering increased knowledge about COVID-19 and increased available medical resources against the virus, including vaccines and medications. Because the current longitudinal study had a relatively long study period (21 months) compared with previous longitudinal studies under the COVID-19 pandemic (less than 17 months based on a recent systematic review and additional literature search) [35,36], the current study was able to detect a relationship between the decrease in psychological distress and fear of COVID-19 and several successive pandemic waves using a longitudinal study design for about 21 months.

The association between occupation and fear of COVID-19 revealed less initial distress and fear among physicians or nurses compared with other hospital staff, also supporting a possibility that resilience was related to increased medical information and/or easy access to medical resources against COVID-19 and/or infectious diseases. Although a number of studies have reported increased distress in relation to the pandemic among nurses [6], a limited number of cross-sectional studies showed relationships between occupation type among healthcare workers and mental health during the pandemic in spite of inconsistencies in results across depression, anxiety, and stress [37]. Professions that were vulnerable to depression included other medical staff and students; doctors, nurses, and students were vulnerable to anxiety; and other

medical staff, students, and financial staff were vulnerable to stress [37]. A meta-analysis of cross-sectional studies showed a tendency for a high prevalence of depression in nurses compared with physicians [5]. The present longitudinal study extends previous findings, suggesting a differential trajectory of mental distress among hospital staff with different occupations: the results indicated that there was consistently less initial distress and fear and subsequently less decrease in mental distress among physicians and nurses compared with other hospital staff members. Doctors and nurses experienced less psychological distress and fear of COVID-19 than individuals in other hospital occupations. It could also be expected that as physicians and nurses have more direct patient contact, repeated exposure to the stress of potential infection might lead to a greater distress tolerance, especially if they have not become ill themselves. This could also be attributed to their comparably easier access to medical information about COVID-19 or their greater ability to take precautions against infection than other hospital workers. Therefore, taken together, the present results support the possibility that some intervention for increasing knowledge and access to medical information about infectious diseases and taking precautions can reduce mental distress in response to a pandemic.

In the current study, high trait anxiety was a predictor of the initial elevation and the subsequent decrease in psychological distress. Previous research has suggested various predisposing factors for higher mental distress during the COVID-19 pandemic, including trait anxiety. Trait anxiety was reported to be associated with perceived vulnerability concerning health and fear of COVID-19 [17]. Students who showed higher trait anxiety were found to express more negative emotions and to perceive themselves as having less academic self-efficacy [38]. Patients with major depression who exhibited stronger suicidal ideation were reported to have higher impulsivity and trait anxiety [39]. The present longitudinal study further suggests the potential of trait anxiety as an easily evaluated predictor of the trajectory of psychological distress in response to stressors.

Some methodological limitations of the current study should be considered. First, because the current study did not include an assessment before the pandemic, the initially elevated trait anxiety, psychological distress, and fear of COVID-19 cannot differentiate between the influence of the pandemic and basal traits. Second, the lower consistency in the number and interval of data collection periods limited the robustness of findings. Third, although the homogeneity of the current study sample is well-suited to exploring individual differences, it could work against learning anything about contextual factors, such as governmental stringency in response to the pandemic or geographic factors. Given that governments worldwide declared different levels of pandemic restrictions, this study was limited in that pandemic conditions in Japan could not be directly compared to those in other nations. Fourth, information on the current survey was provided to candidates for participation via handouts and posters in each hospital. However, during the pandemic many medical staff left, started another job, or transferred to another department each month. For these reasons, it was difficult to accurately count the total number of candidates among the medical staff in each hospital each week. Therefore, in the current study an accurate response rate could not be calculated, nor could potential selection bias be completely eliminated. Fifth, it is possible that individuals changed their email addresses during the survey period, allowing for the assumption that another person had answered when in fact the same individual responded repeatedly. In this case, duplicate responses could have occurred and might have influenced the current results. Sixth, the group that scored very high on trait anxiety also had high psychological distress and fear of COVID-19 scores. However, this result could be a statistical phenomenon as well as a psychological phenomenon. Seventh, although health attendants and environmental services workers were not included in the current study, previous studies showed that they experience more significant psychological stress than medical staff [40]. Therefore, future studies should include

them to further clarify the psychological differentiation among occupational groups. Eighth, in the current study, the variation in socioeconomic status of medical staff during the pandemic was not considered because of a lack of sufficient data on resignation rates and wage changes during the study period.

## Conclusions

The current longitudinal study in a relatively homogeneous sample supports the possibility that increased accessibility of medical information and resources decreased psychological distress and fear regarding COVID-19 and the usefulness of trait anxiety measurement and occupation as a predictor of mental health outcomes.

## Supporting information

**S1 Fig. Weekly average number of newly infected COVID-19 cases and K6 or FCV-19S scores.** Weekly average number of newly infected COVID-19 cases and K6 (A) or FCV-19S (B) scores were shown during the observation period.
(TIF)

**S1 Table. Effects of time and trait anxiety or occupation type on K6 scores for 100 weeks[†] or earlier and after 100 weeks.** [†]The day of the COVID-19 outbreak in Japan is defined as "week 1" (January 16, 2020). [*]Statistical significance after Bonferroni correction (p < 0.0167). Abbreviations: B, regression coefficient; 95% CI, 95% confidence interval.
(DOCX)

**S2 Table. Effects of time and trait anxiety or occupation type on FCV-19S scores for 100 weeks[†] or earlier and after 100 weeks.** [†]The day of the COVID-19 outbreak in Japan is defined as "week 1" (January 16, 2020). [*]Statistical significance after Bonferroni correction (p < 0.0167). Abbreviations: B, regression coefficient; 95% CI, 95% confidence interval.
(DOCX)

**S1 File.**
(XLSX)

**S2 File.**
(XLSX)

## Acknowledgments

We express our gratitude to all study participants and to the staff at the hospitals that participated in this survey; Fujieda Municipal General Hospital, Fuyo association Seirei Numazu Hospital, Hamamatsu Medical Center, Ito municipal hospital, Iwata city Hospital, JA Shizuoka Kohseiren Enshu Hospital, Kikugawa General Hospital, Kiseikai Shin-fuji Hospital, Koka community Hospital, Numazu City Hospital, Seirei Hamamatsu General Hospital, Seirei Mikatahara General Hospital, Shimada General Medical Center, Shizuoka Children's Hospital, Shizuoka City Hospital, Shizuoka Saiseikai General Hospital, and Susono Red Cross Hospital. We thank Benjamin Knight, MSc., from Edanz (https://jp.edanz.com/ac) for editing a draft of this manuscript.

## Author Contributions

**Conceptualization:** Yosuke Kameno, Noriyuki Enomoto, Hidenori Yamasue.

**Data curation:** Yosuke Kameno, Yumi Naito.

**Formal analysis:** Tomoko Nishimura, Hidenori Yamasue.

**Investigation:** Yosuke Kameno, Yumi Naito, Daisuke Asai, Jun Inoue, Yosuke Mochizuki, Tomoyo Isobe, Atsuko Hanada, Noriyuki Enomoto.

**Methodology:** Yosuke Kameno, Tomoko Nishimura, Hidenori Yamasue.

**Project administration:** Yosuke Kameno, Hidenori Yamasue.

**Resources:** Yosuke Kameno, Hidenori Yamasue.

**Supervision:** Hidenori Yamasue.

**Validation:** Yosuke Kameno, Tomoko Nishimura, Yumi Naito, Daisuke Asai, Jun Inoue, Yosuke Mochizuki, Tomoyo Isobe, Atsuko Hanada, Noriyuki Enomoto, Hidenori Yamasue.

**Visualization:** Yosuke Kameno, Tomoko Nishimura, Hidenori Yamasue.

**Writing – original draft:** Yosuke Kameno, Tomoko Nishimura, Hidenori Yamasue.

**Writing – review & editing:** Yosuke Kameno, Tomoko Nishimura, Hidenori Yamasue.

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
