## [Decision Letter · Decision Letter 0]

5 Apr 2023

PONE-D-23-03982Time-course changes in mental distress and their predictors in response to the coronavirus disease 2019 (COVID-19) pandemic: A long-term longitudinal multi-site study of hospital staffPLOS ONE

Dear Dr. Yamasue,

Thank you for submitting your manuscript to PLOS ONE. After careful consideration, we feel that it has merit but does not fully meet PLOS ONE’s publication criteria as it currently stands. Therefore, we invite you to submit a revised version of the manuscript that addresses the points raised during the review process.

We look forward to receiving your revised manuscript.

Kind regards,

Qin Xiang Ng, MD, MPH

Academic Editor

PLOS ONE

Journal Requirements:

Reviewers' comments:

Reviewer's Responses to Questions

**Comments to the Author**

1. Is the manuscript technically sound, and do the data support the conclusions?

Reviewer #1: Yes

Reviewer #2: Yes

Reviewer #3: Partly

Reviewer #4: Partly

2. Has the statistical analysis been performed appropriately and rigorously? 

Reviewer #1: No

Reviewer #2: No

Reviewer #3: I Don't Know

Reviewer #4: N/A

3. Have the authors made all data underlying the findings in their manuscript fully available?

Reviewer #1: Yes

Reviewer #2: Yes

Reviewer #3: Yes

Reviewer #4: Yes

4. Is the manuscript presented in an intelligible fashion and written in standard English?

Reviewer #1: Yes

Reviewer #2: Yes

Reviewer #3: No

Reviewer #4: Yes

5. Review Comments to the Author

Reviewer #1: PLOS One manuscript PONE-D-23-03982, entitled "Time-course changes in mental distress and their predictors in response to the coronavirus disease 2019 (COVID-19) pandemic: A long-term longitudinal multi-site study of hospital staff", is an empirical research report (N = 545 staff members at 18 hospitals treating COVID-19 patients in Shizuoka Prefecture, Japan), of individual differences in the longitudinal trajectory of mental distress over 21 months of COVID-19 waves in Japan. Psychological distress and fear of COVID-19 were initially elevated but then decreased over the course of the investigation, especially for those high in trait anxiety and for specific occupations. The time course of observed effects was also examined with cross-correlation analyses.

This is a well-written manuscript on a topic of substantial interest to the readership of PLOS One. The data are quite valuable and were capably analyzed. The results were somewhat countER-intuitive, but the authors have done a good job explaining them.

The Introduction appropriately motivates the study and embeds the work in a scholarly and contemporary review of the relevant literature. The authors' remarks about temporal resolution were especially useful.

1. The psychometric properties of the primary study instruments, K6 and FCV-19S, are not reported.

2. As the authors point out, the homogeneity of the sample is well suited to exploring individual differences. However, it works against learning anything about contextual factors, such as governmental stringency in response to the pandemic and geographic factors. This is not a defect in the study itself, but it is a constraint on the scope of the questions that might be asked about this topic. The Discussion might address this design feature.

3. For the cross-correlation analyses the authors created weekly averages. First of all, were these moving averages? Second, if the causal relationship between infections and distress / fear operates over an interval shorter than a week, won't the effect be obscured by the weekly averaging? This was a bit of a surprise in an article that seemed especially attuned to temporal resolution issues. If the binning of observations was to make "lags" meaningful and comparable, the authors should say so. Similarly, the division of the observation period into only two periods and the use of the somewhat arbitrary 100 week cut-off for this division both seem rather coarse from a measurement perspective (vs., perhaps, a cubic or other polynomial model, with anxiety as a continuous moderator). Also, re-running models within categories of interest (e.g., very high versus high anxiety or doctors versus nurses or >100 weeks versus <=100 weeks) and comparing significance levels is not the same as testing interactions, which could be done more directly (e.g., Cohen, Cohen, West, & Aiken, 2003, 3rd ed.).

4. Line 263, the lag 0 cross-correlation should be a POSITIVE +.27.

5. A *partial* cross-correlation function may expose the effect the authors are seeking more clearly. The apparent smooth periodicity of the CCF may simply be the effects of autocorrelation. The PACF can control for this, leaving only the spikes in effects across lags.

6. Line 314: Would the authors care to elaborate, even briefly on what they mean by, "The present results further support the possibility that increasing knowledge and access to medical information about infectious diseases can reduce mental distress in response to 316 a pandemic."

Reviewer #2: The present study tested the effects of COVID-19 waves on mental health among medical staffs at 18 hospitals treating COVID-19 patients in Japan. Authors found that contrary to increasing new infected cases as waves progressed, initially elevated psychological distress (K6) and fear of COVID-19 (FCV-19S) were decreased among waves. These trends were more prominent in individuals with high trait anxiety and in occupations other than physicians or nurses.

This study is meaningful and interesting to examine longitudinal mental health data of medical healthcare workers for 21 months including COVID-19 pandemic waves. I suggest the following points to improve the manuscript.

1. Authors should state the response rate of the participants in each hospital to discuss sample bias.

2. To evaluate effects of COVID-19 waves on mental health among medical staff, their experience of work in the COVID-19 unit or suffering from the illness are more important rather than their baseline anxiety and occupation. Why did authors not examine them?

3. I think data of K6 and FCV do not exhibit a normal distribution. Therefore, Generalized Linear Mixed Model may be more suitable than LMM for statistical analyses in such a case.

4. Cross correlation analysis make the central theme of the manuscript vague. Cross-correlation r values are relatively small. I wonder if these analyses are really necessary.

Reviewer #3: The primary finding of this longitudinal study that anxiety/distress and fear of COVID increased early in the pandemic and then decreased with further exposure to the pandemic is worth presenting, but the report is much longer than is warranted by the finding. I would suggest greatly condensing all sections of the paper, focusing on the primary point. The suggestion about the duration of follow-up of staff may be less important than a recommendation about when and what kind of intervention might improve health care worker adaptation to pandemics. You might consider as an additional explanation of your findings that physicians and nurses have more direct patient contact, so that repeated exposure to the stress of potential infection, especially without getting sick oneself, leads to tolerance to distress rather than intellectual mastery of it. The impact of high trait anxiety could be a statistical as well as a psychological phenomenon. Finally, you might clarify whether you corrected statistically for multiple comparisons.

Reviewer #4: I read this manuscript with interest, however, the report is difficult to follow and several methodological issues exist:

1. Some comments on the local COVID-19 situation and policies on the ground during the period the data was conducted would be helpful as well.

2. Would the psychometric scores be high because of the generally high amount of uncertainty and fear during the early stages of the pandemic? How do we account for this?

3. More details are necessary on the dissemination and sampling process. Did the authors take steps to guard against duplicate responses to the survey?

4. “The ethical committee of Hamamatsu University School of Medicine approved this study protocol as a retrospective study on 29 March 2022 and informed consent was obtained from participants in the form of an opportunity to opt out on the website (20-083)” - does this mean the study was approved retrospectively rather than prospectively? Unclear why this is the case. It seems the authors did not obtain the necessary ethical approval before embarking on the study, especially for a multi-site study.

5. “The current long-term longitudinal study” - redundant to say “long-term longitudinal”.

6. “… demographics at the basal assessment” - please change “basal” to “initial”.

7. Under the occupational groups, health attendants and environmental services workers appear to be left out. This should be an area for future research as previous studies have found that they experience significant (if not greater) psychological stress than medical staff (citation: pubmed.ncbi.nlm.nih.gov/36116538).

8. Other study limitations were not discussed. In this panel survey study, it is hard to tell exactly who responded to the study and the individuals surveyed at the different time points may not be the same person. Also, what about the likelihood of selection bias (of unknown extent because attrition cannot be evaluated)?

9. Given the economic impact of the pandemic, socioeconomic status may still vary over time amongst the individuals sampled, which was not assessed in the present study. This is a study limitation as well.

10. “.. desirable period in which to follow up high-risk individuals after exposure to stressors (e.g. from 0 to 7 weeks)” - how was this derived? This is unclear.

6. PLOS authors have the option to publish the peer review history of their article (what does this mean?). If published, this will include your full peer review and any attached files.

Reviewer #1: No

Reviewer #2: No

Reviewer #3: No

Reviewer #4: No

---

## [Author Response · Author response to Decision Letter 0]

2 Jun 2023

1st June 2023

Dr. Emily Chenette, PhD, and Dr. George Vousden, PhD

Editor-in-Chief, PLOS ONE 

Dear Dr. Chenette and Dr. Vousden,

Thank you for your kind letter on April 6, 2023, regarding our manuscript PONE-D-23-03982 entitled “Time-course changes in mental distress and their predictors in response to the coronavirus disease 2019 (COVID-19) pandemic: A long-term longitudinal multi-site study of hospital staff”. We greatly appreciate the comments and suggestions provided by the editor and reviewer for the revision of our manuscript.

We have revised our manuscript in accordance with the suggestions of the reviewer to reword some parts of manuscript for clarity or to improve statistical analyses. Our point-by-point responses to all of the comments are presented in the response letter; reviewer comments are in italics with our responses immediately following each comment. The revised parts of the text are underlined in the re-revised manuscript. Please see the responses in the "PONE-D-23-03982R_response.docx".

We believe that our manuscript has been improved by incorporating the helpful comments of the reviewer and deeply appreciate the valuable input. Please do not hesitate to contact us if any further information is required.

Sincerely,

Hidenori Yamasue, MD, PhD

Department of Psychiatry, Hamamatsu University School of Medicine 

1-20-1 Handayama, Higashiku, Hamamatsu City 431-3192, Japan

Phone: +81-53-435-2295; Fax: +81-53-435-3621; E-mail: yamasue@hama-med.ac.jp

We recommend the following four reviewers for this manuscript:

---

## [Decision Letter · Decision Letter 1]

27 Jul 2023

PONE-D-23-03982R1Time-course changes in mental distress and their predictors in response to the coronavirus disease 2019 (COVID-19) pandemic: A longitudinal multi-site study of hospital staffPLOS ONE

Dear Dr. Yamasue,

Thank you for submitting your manuscript to PLOS ONE. After careful consideration, we feel that it has merit but does not fully meet PLOS ONE’s publication criteria as it currently stands. Therefore, we invite you to submit a revised version of the manuscript that addresses the points raised during the review process.

We look forward to receiving your revised manuscript.

Kind regards,

Qin Xiang Ng, MD, MPH

Academic Editor

PLOS ONE

Journal Requirements:

Reviewers' comments:

Reviewer's Responses to Questions

**Comments to the Author**

1. If the authors have adequately addressed your comments raised in a previous round of review and you feel that this manuscript is now acceptable for publication, you may indicate that here to bypass the “Comments to the Author” section, enter your conflict of interest statement in the “Confidential to Editor” section, and submit your "Accept" recommendation.

Reviewer #1: (No Response)

Reviewer #3: (No Response)

2. Is the manuscript technically sound, and do the data support the conclusions?

Reviewer #1: Partly

Reviewer #3: Partly

3. Has the statistical analysis been performed appropriately and rigorously? 

Reviewer #1: No

Reviewer #3: No

4. Have the authors made all data underlying the findings in their manuscript fully available?

Reviewer #1: Yes

Reviewer #3: Yes

5. Is the manuscript presented in an intelligible fashion and written in standard English?

Reviewer #1: Yes

Reviewer #3: No

6. Review Comments to the Author

Reviewer #1: Manuscript PONE-D-23-03982R1 entitled “Time-course changes in mental distress and their predictors in response to the coronavirus disease 2019 (COVID-19) pandemic: A long-term longitudinal multi-site study of hospital staff" is the first revision of an initial submission upon which I served as Reviewer 1.

In this revision, the authors have done a good job responding to my initial concerns. Some remaining issues include:

1. While reliability information is welcome in these two places, it could be made slightly more clear that these reliability coefficients come from other studies, not the current study. Also, it would be nice to get an additional remark about the validity of the K6 (viz., that it correlates with other measures or ratings of mental distress?), as was done for the FCV-19S.

3. Firstly, I remain unconvinced that the GLMM and cross-correlation models address the different measurement intervals problem. Once you've binned the data by week, the different measurement intervals are lost, right? Every participant's data are one week at a time, regardless of where in that week the observation(s) occurred. GLMM and DCCA are capable of addressing the variable intervals, but that problem is solved by binning the data by week. If the authors agree, then I'd suggest simply deleting the sentence, "However, the employed generalized linear mixed model allows for different measurement times and intervals for each individual, and detrended cross- correlation analysis is considered to be less influenced by these limitations."

Second, the cross-correlation analyses remain unclear. In my review I requested the partial cross-correlation functions (PCCF), but then I referred to these as "PACF." In their reply, the authors have provided a partial autocorrelation function for each variable, presumably in response to the erroneous abbreviation in my third sentence, rather than the partial CROSS-correlation I referred to in my first sentence. I apologize for the inconsistency in my review, and I would now like to invite the authors to see whether the cross correlations they report would change by correcting them for autocorrelation within the variables. Conceptually, what I am asking them to look at is the extent to which autocorrelation accounts for the apparent lagged effects. The partial cross correlation corrects the lagged effects for the covariance accounted for by the concurrent and intervening effects. The authors have picked the peak in a smooth autocorrelation function to claim special status for, in the case of the K6, the 7 week lag. I just don't think this is supportable from these analyses. In short, the cross-correlation function needs to be interpreted in light of the partial cross-correlation function (PCCF) that corrects for autocorrelation and intervening effects.

That said, like Reviewer 2 (point #4), I do not view the lagged effects as crucial to the contribution of the manuscript, so if the authors end up electing to remove the cross correlation analyses and discussion entirely it would not alter my evaluation of the manuscript. The PCCF could, however, be quite interesting.

Reviewer #3: You have answered many of the issues raised by reviewers, but the manuscript is still very long considering the information provided. The recommendation that as staff encounter more cases they will become desensitized if they do not become ill themselves is likely to be correct, but you still might add recommendations for specific interventions, or state that no intervention is necessary.

7. PLOS authors have the option to publish the peer review history of their article (what does this mean?). If published, this will include your full peer review and any attached files.

Reviewer #1: No

Reviewer #3: No

---

## [Author Response · Author response to Decision Letter 1]

20 Aug 2023

21st August 2023

Dr. Emily Chenette, PhD, and Dr. George Vousden, PhD

Editor-in-Chief, PLOS ONE 

Dear Dr. Chenette and Dr. Vousden,

Thank you for your kind letter on 28th July 2023, regarding our manuscript PONE-D-23-03982 entitled “Time-course changes in mental distress and their predictors in response to the coronavirus disease 2019 (COVID-19) pandemic: A long-term longitudinal multi-site study of hospital staff.” We greatly appreciate the comments and suggestions provided by the editor and reviewers for the re-revision of our manuscript.

We have substantially revised our manuscript in accordance with your suggestions, as well as those of the reviewers. Our point-by-point responses to all of the comments are presented below; reviewer comments are in italics with our responses immediately following each comment. The revised sections of the text are underlined in each response as well as in the revised manuscript itself. The line numbers in the responses were based on the marked up copy version of revised manuscript.

Reviewer: 1

In this revision, the authors have done a good job responding to my initial concerns.

We appreciate the reviewer’s kind and careful re-review.

1) While reliability information is welcome in these two places, it could be made slightly more clear that these reliability coefficients come from other studies, not the current study. Also, it would be nice to get an additional remark about the validity of the K6 (viz., that it correlates with other measures or ratings of mental distress?), as was done for the FCV-19S.

We thank the reviewer for giving us the opportunity to clarify this point. As suggested, the following sentences were added to the revised manuscript: 

Methods:

Lines 114-118 of the revised manuscript: “A previous study reported that Cronbach’s α of the Japanese version of K6 scale was 0.85 [14]. The screening performance of this scale was comparable with that of Center for Epidemiologic Studies-Depression Scale (CES-D), and better than that of Depression and Suicide Screen (DSS) in the Japanese version [14].”

Lines 121-125 of the revised manuscript: “A previous study also revealed Cronbach’s α of the Japanese version of FCV-19S scale was 0.82 [9]. This scale showed a significant positive correlation to Generalized Anxiety Disorder (GAD)-7, Patient Health Questionnaire for Adolescents (PHQ-A), and Perceived Vulnerability to Disease Scale (PVDS) in the Japanese version. The scale had sufficiently high internal confidence and moderately good construct validity. [9]”

2) Firstly, I remain unconvinced that the GLMM and cross-correlation models address the different measurement intervals problem. Once you've binned the data by week, the different measurement intervals are lost, right? Every participant's data are one week at a time, regardless of where in that week the observation(s) occurred. GLMM and DCCA are capable of addressing the variable intervals, but that problem is solved by binning the data by week. If the authors agree, then I'd suggest simply deleting the sentence, "However, the employed generalized linear mixed model allows for different measurement times and intervals for each individual, and detrended cross-correlation analysis is considered to be less influenced by these limitations."

We thank the reviewer for this insightful comment. As suggested, the following sentences were deleted from the revised manuscript: 

Discussion:

Lines 400-402 of the revised manuscript: "However, the employed generalized linear mixed model allows for different measurement times and intervals for each individual, and detrended cross-correlation analysis is considered to be less influenced by these limitations."

Second, the cross-correlation analyses remain unclear. In my review I requested the partial cross-correlation functions (PCCF), but then I referred to these as "PACF." In their reply, the authors have provided a partial autocorrelation function for each variable, presumably in response to the erroneous abbreviation in my third sentence, rather than the partial CROSS-correlation I referred to in my first sentence. I apologize for the inconsistency in my review, and I would now like to invite the authors to see whether the cross correlations they report would change by correcting them for autocorrelation within the variables. Conceptually, what I am asking them to look at is the extent to which autocorrelation accounts for the apparent lagged effects. The partial cross correlation corrects the lagged effects for the covariance accounted for by the concurrent and intervening effects. The authors have picked the peak in a smooth autocorrelation function to claim special status for, in the case of the K6, the 7 week lag. I just don't think this is supportable from these analyses. In short, the cross-correlation function needs to be interpreted in light of the partial cross-correlation function (PCCF) that corrects for autocorrelation and intervening effects.

That said, like Reviewer 2 (point #4), I do not view the lagged effects as crucial to the contribution of the manuscript, so if the authors end up electing to remove the cross correlation analyses and discussion entirely it would not alter my evaluation of the manuscript. The PCCF could, however, be quite interesting.

We appreciate the reviewer’s insightful and important comments. We agreed with the comments and also thought that it would be rather important to focus on the time-course changes and the association between the outcomes and anxiety and occupation resulting from GLMM analysis than to emphasize the result of cross-correlation analysis. Therefore, we removed and revised the sentence and figures from the revised manuscript related to the result and discussion of cross-correlation analyses as follows.

Abstract:

Lines 33-36 of the revised manuscript: ”Cross-correlation analyses demonstrated that K6 and FCV-19S exhibited time-course changes, with increases in these indices occurring simultaneously with an increase in infections and decreases delayed by 7 weeks after an increase in infections.”

Lines 38-39 of the revised manuscript: “The results also indicated a desirable period for following up high-risk individuals after exposure to stressors.”

Introduction:

Lines 62-66 of the revised manuscript: “ Furthermore, high temporal resolution is necessary to test the relationships between mental distress, such as fear or anxiety, and the onset of stressors, because previous studies have indicated that these relationships operate at the scale of days to weeks [8]. However, the temporal resolution of most previous longitudinal research on COVID-19 mental health has been limited to months to years, with a limited number of time-points [9].”

Lines 74-76 of the revised manuscript: ”Additionally, using a high temporal resolution analysis of long-term longitudinal data, the time-course changes in the relationship between the pandemic waves and mental distress were also examined.“

Methods:

Lines 184-194 of the revised manuscript: “Next, to examine how the number of newly infected cases of COVID-19 was correlated with psychological distress and fear of the COVID-19, respectively, we conducted cross-correlation analyses. Cross-correlation analysis allows us to understand the similarity of data in a time series and the lag of the period. The cross-correlation between the number of newly infected cases in the prefecture and the scores for psychological distress / fear of COVID-19 were all calculated as weekly averages. We utilized weekly averages in the analyses, because mental health surveys were not completed by each participant on each day; therefore, when the unit of time was set to one day, the amount of missing data was very large. Furthermore, because the number of newly infected cases tended to increase, and because psychological distress and fear of COVID-19 could also have a trend through the period, we conducted a detrended cross-correlation analysis (DCCA) using standardized values.”

Lines 195-196 of the revised manuscript: “Additionally, the observation period was divided into two periods (before and after 100 weeks), and a GLMM and DCCA was then conducted for each period.”

Lines 200-201 of the revised manuscript: “GLMM was conducted using STATA version 17.0 [22]. and DCCA was conducted using Python version 3.6.

Result:

Lines 286-311 of the revised manuscript: we removed the section of the result of cross-correlation analysis.

Discussion:

Lines 323-327 of the revised manuscript: “Focusing on each pandemic wave, psychological distress and fear of COVID-19 also showed time-course changes as waves with increases that were delayed by 0–2 weeks after an increase in the number of infections and decreases that were delayed by 7 weeks after an increase in the number of infections. The wave of fear regarding COVID-19 tended to be slightly delayed after the wave of psychological distress.”

Lines 378-391 of the revised manuscript: “The present study showed time-course changes in psychological distress and the fear of COVID-19 with increases in the distress and fear delayed by 0–2 weeks after an increase in the number of infections and decreases in the distress and fear delayed by 7 weeks after an increase in the number of infections. Previous research on the effects of acute stressors on anxiety and psychological reactions in various populations have shown responses to stressors within a few weeks [43]. Previous longitudinal studies with weekly measurement under the COVID-19 pandemic were limited. A large-scale 20-week longitudinal study from March to August 2020 reported that anxiety and depression levels both declined across the first 20 weeks following the introduction of lockdown in England, suggesting an adaptation for the situation. The fastest decreases were observed across the strict lockdown period (i.e., 2 to 5 weeks), with symptoms plateauing as further lockdown easing measures were shown (16 to 20 weeks) [44]. The present results across pandemic waves revealed the peak and subsequent decrease of distress and fear after a stressor extended current knowledge regarding the desirable timeframe for mental health care under the onset of stressors such as a pandemic of infectious disease, as well as other disasters.”

Lines 396-402 of the revised manuscript: “To reduce the potential confounding effect, weekly averages were calculated to summarize the observations. These weekly averages could make the temporal resolution insufficient, as the lag between the peak of increase in the number of infections and the peaks of psychological distress and fear of COVID-19 could have occurred at intervals shorter than one week. However, the employed generalized linear mixed model allows for different measurement times and intervals for each individual, and detrended cross-correlation analysis is considered to be less influenced by these limitations.”

Conclusion:

Lines 429-431 of the revised manuscript: “Furthermore, the current study suggests a desirable period in which to follow up high-risk individuals after exposure to stressors (e.g., from 0 to 7 weeks; based on the results of current cross-correlation analyses showing the timing of elevations of fear and anxiety).”

Supporting information:

Lines 608-614 of the revised manuscript: “S2 Fig. Cross-correlation between weekly average number of newly infected COVID-19 cases and K6 score for (A) 100 weeks† or earlier and (B) after 100 weeks. †The day of the COVID-19 outbreak in Japan is defined as “week 1” (January 16, 2020).

S3 Fig. Cross-correlation between weekly average number of newly infected COVID-19 cases and FCV-19S score for (A) 100† weeks or earlier and (B) after 100 weeks. †The day of the COVID-19 outbreak in Japan is defined as “week 1” (January 16, 2020).”

Figures:

We removed Fig 3, S2 Fig, and S3 Fig.

Reviewer: 3

You have answered many of the issues raised by reviewers, but the manuscript is still very long considering the information provided. 

We appreciate the reviewer’s helpful comment. Together with the suggestions by Reviewer 1, we removed all the sentences and figures related to cross-correlation analysis, and these revision made the manuscript more concise. The main text of re-revised manuscript is 4,003words, while that of revised manuscript was 4,883 words. 

The recommendation that as staff encounter more cases they will become desensitized if they do not become ill themselves is likely to be correct, but you still might add recommendations for specific interventions, or state that no intervention is necessary.

Thank you for giving us this opportunity to further clarify the issue. e have revised the text regarding the intervention as follows:

Line 360-368 of the revised manuscript: “Doctors and nurses experienced less psychological distress and fear of COVID-19 than individuals in other hospital occupations. It could also be expected that as physicians and nurses have more direct patient contact, repeated exposure to the stress of potential infection might lead to a greater distress tolerance, especially if they have not become ill themselves. This could also be attributed to their comparably easier access to medical information about COVID-19 or their greater ability to take precautions against infection than other hospital workers. Therefore, taken together, the present results support the possibility that some intervention for increasing knowledge and access to medical information about infectious diseases and taking precautions can reduce mental distress in response to a pandemic.”

In addition, we also carefully check the current manuscript and revised the sentences as follows:

In the abstract, the statistical results were added for the FCV-19S as follows:

Lines 28-31 of the revised manuscript: “Contrary to increasing new infected cases as waves progressed, initially elevated psychological distress (K6) and fear of COVID-19 (FCV-19S) were decreased among waves (K6: B = -.02, 95% confidence interval [CI] = -.03 to -.01; FCV-19S: B = -.10, 95% CI = -.16 to -.04).”

In the Table 1, the layout was corrected and “K6” was added as follows:

Row 2: “Outcome measures at the initial assessment 545 6.01 5.29 0 24”

Row 3: “K6 545 6.01 5.29 0 24”

In the Supporting information, the title of supplemental tables were added as follows:

S1 Table: “S1 Table. Effects of time and trait anxiety or occupation type on K6 scores for 100 weeks or earlier and after 100 weeks”

S2 Table: “S2 Table. Effects of time and trait anxiety or occupation type on FCV-19S scores for 100 weeks or earlier and after 100 weeks”

---

## [Decision Letter · Decision Letter 2]

19 Sep 2023

Time-course changes in mental distress and their predictors in response to the coronavirus disease 2019 (COVID-19) pandemic: A longitudinal multi-site study of hospital staff

PONE-D-23-03982R2

Dear Dr. Yamasue,

We’re pleased to inform you that your manuscript has been judged scientifically suitable for publication and will be formally accepted for publication once it meets all outstanding technical requirements.

Kind regards,

Qin Xiang Ng, MBBS, GDMH, MPH

Academic Editor

PLOS ONE

Additional Editor Comments (optional):

Reviewers' comments:

Reviewer's Responses to Questions

**Comments to the Author**

1. If the authors have adequately addressed your comments raised in a previous round of review and you feel that this manuscript is now acceptable for publication, you may indicate that here to bypass the “Comments to the Author” section, enter your conflict of interest statement in the “Confidential to Editor” section, and submit your "Accept" recommendation.

Reviewer #1: All comments have been addressed

Reviewer #3: (No Response)

2. Is the manuscript technically sound, and do the data support the conclusions?

Reviewer #1: (No Response)

Reviewer #3: Partly

3. Has the statistical analysis been performed appropriately and rigorously? 

Reviewer #1: (No Response)

Reviewer #3: I Don't Know

4. Have the authors made all data underlying the findings in their manuscript fully available?

Reviewer #1: (No Response)

Reviewer #3: Yes

5. Is the manuscript presented in an intelligible fashion and written in standard English?

Reviewer #1: (No Response)

Reviewer #3: Yes

6. Review Comments to the Author

Reviewer #1: (No Response)

Reviewer #3: You have provided comprehensive replies to the reviewers' comments. Since this manuscript will not be copy edited, the remaining redundant language, for example in the descriptions of the STAI, should be eliminated through careful editing of the manuscript. You might also consider the possibility that nurses and doctors are more directly familiar with serious illness, including communicable diseases, allowing them to accomodate more rapidly to continued exposure to COVID without becoming ill themselves. You could also emphasize the point that you have no specific recommendations for reducing emotional responses to pandemics in health care workers. There may be differences in cultural responses to this kind of stress in addition to differences in how governments responded to the pandemic.

7. PLOS authors have the option to publish the peer review history of their article (what does this mean?). If published, this will include your full peer review and any attached files.

Reviewer #1: No

Reviewer #3: No

---

## [Editor Report · Acceptance letter]

26 Sep 2023

PONE-D-23-03982R2 

Time-course changes in mental distress and their predictors in response to the coronavirus disease 2019 (COVID-19) pandemic: A longitudinal multi-site study of hospital staff 

Dear Dr. Yamasue:

I'm pleased to inform you that your manuscript has been deemed suitable for publication in PLOS ONE. Congratulations! Your manuscript is now with our production department. 

Kind regards, 

on behalf of

Dr. Qin Xiang Ng 

Academic Editor

PLOS ONE